# Therapeutic Significance of NLRP3 Inflammasome in Cancer: Friend or Foe?

**DOI:** 10.3390/ijms252413689

**Published:** 2024-12-21

**Authors:** Aliea M. Jalali, Kenyon J. Mitchell, Christian Pompoco, Sudeep Poludasu, Sabrina Tran, Kota V. Ramana

**Affiliations:** Department of Biomedical Sciences, Noorda College of Osteopathic Medicine, Provo, UT 84606, USA

**Keywords:** cancer, NLRP3, inflammasome, innate immunity, IL-1b, caspase-1

## Abstract

Besides various infectious and inflammatory complications, recent studies also indicated the significance of NLRP3 inflammasome in cancer progression and therapy. NLRP3-mediated immune response and pyroptosis could be helpful or harmful in the progression of cancer, and also depend on the nature of the tumor microenvironment. The activation of NLRP3 inflammasome could increase immune surveillance and the efficacy of immunotherapy. It can also lead to the removal of tumor cells by the recruitment of phagocytic macrophages, T-lymphocytes, and other immune cells to the tumor site. On the other hand, NLRP3 activation can also be harmful, as chronic inflammation driven by NLRP3 supports tumor progression by creating an environment that facilitates cancer cell proliferation, migration, invasion, and metastasis. The release of pro-inflammatory cytokines such as IL-1β and IL-18 can promote tumor growth and angiogenesis, while sustained inflammation may lead to immune suppression, hindering effective anti-tumor responses. In this review article, we discuss the role of NLRP3 inflammasome-mediated inflammatory response in the pathophysiology of various cancer types; understanding this role is essential for the development of innovative therapeutic strategies for cancer growth and spread.

## 1. Introduction

The NOD-like receptor (NLR) protein family, a type of pattern recognition receptor (PRR), plays a vital role in the cellular innate immune response by identifying foreign substances, triggering an immune response, and assembling inflammasomes [1]. Within humans, there are currently 14 known NLR protein members, named NLPR1 through NLPR14, involved in various aspects of immune cell regulation [1]. Activation of NLR proteins leads to inflammatory responses mediated through nuclear factor kappa binding protein (NF-κB), mitogen activated protein kinase (MAPK), or caspase-1 activation and the release of proinflammatory cytokines [2]. Dysregulation and malfunction of NLR variants have been linked with the pathophysiology of inflammatory diseases, and there is also evidence that they may contribute to cancer progression by controlling inflammatory signaling pathways to promote cell proliferation, migration, and invasion [3,4].

The Nod-like receptor protein 3 (NLRP3) inflammasome is a multi-protein assembly that triggers initial innate immune responses by activating caspase-1 and releasing proinflammatory cytokines such as interleukin (IL)-1β and IL-18 in response to pathogen-associated molecular patterns (PAMPs) and damage-associated molecular patterns (DAMPs) (Figure 1) [5]. The inflammasome activation could cause a specific programmed cell death called pyroptosis. Generally, pyroptosis helps to eliminate infected or damaged cells. Thus, inflammasomes are essential for defending the body against pathogen attacks. However, their overactivation or dysfunction is linked to various inflammatory diseases, including gout, atherosclerosis, and cancer. The NLRP3 inflammasome is of particular interest, due to its ability to have both protective and damaging effects on carcinogenesis, depending on the type of cancer [6]. Further, several studies have indicated that overactivation of the NLRP3 inflammasome has been linked to cancer progression [1,4,7]. Conversely, deficiencies in the NLRP3 inflammasome have shown a potential to increase tumor burden in certain cancers, such as colitis-associated cancer, due to a lack of tumor-suppressing cytokines [6,7].

Abderrazak et al. and Leemans et al. [8,9] have demonstrated that NLPR3 inflammasome-generated IL-1β and IL-18 promote immune and inflammatory responses and accelerate tumor growth and spread. IL-1β and IL-18, generated by the NLRP3-inflammasome, could cause persistent immune and inflammatory responses by activating the generation of various other immune cell-mediated cytokines and chemokines such as tumor necrosis factor-alpha (TNF-α), IL-6, IL-4, interferon-gamma (IFN-γ), monocyte chemoattractant protein-1 (MCP-1) and melanocyte-inhibiting factor (MIF-1) leading to tumor growth and metastasis. Further, this inflammatory state can create a tumor-promoting environment by stimulating cancer cell growth and survival. In cancers such as colon and breast, the inflammatory mediators released by the NLRP3 inflammasome have been associated with increased cancerous growth and metastasis [10,11]. In some instances, IL-1β and IL-18 also contribute to the survival of the cancer cells, and that helps them spread to other organs by modulating the epithelial-to-mesenchymal transition (EMT) [12,13,14]. Therefore, inflammation is a significant risk factor for cancer development in several ways. It can cause DNA damage leading to mutations, disturb the immune balance and promote cell growth, increase the expression of growth factors that initiate angiogenesis required for tumor growth, and impair immune surveillance. Thus, chronic and uncontrolled inflammation caused by NLRP3-dependent and independent mechanisms could lead to many cancer types.

However, the NLRP3 inflammasome can also play a role in suppressing cancer through mechanisms related to immune surveillance. Further, activation of inflammasome could recruit immune cells to the tumor site, leading to the destruction of cancer cells. For instance, Wu et al. [15] have shown that the NLRP3 inflammasome can induce pyroptosis, a programmed cell death that could halt tumor growth. In addition, the inflammatory response triggered by the NLRP3 inflammasome can also stimulate anti-tumor immunity, which enhances the T-cell-mediated elimination of cancer cells [16].

The dual action of the NLRP3 inflammasome in cancer underscores its complex and context-dependent role. On one hand, NLRP3 inflammasome could act as a driver of tumor progression through chronic inflammation. On the other hand, it could support the tumor-associated immune system to promote tumor growth [7,17]. Therefore, further investigations are required to understand the mechanisms through which the NLRP3 inflammasome regulates cancer growth and progression. In this review, we discuss the possible role of NLRP3 inflammasome in the mediation of tumor growth and spread and its potential therapeutic use. We searched PubMed and Google Scholar to find articles published in the last 10 years or so, using keywords including NLRP3, inflammasome, immune response, IL-1β, IL-18, pyroptosis, melanoma, leukemia, breast cancer, colon cancer, lung cancer, and other cancer types. We included various research articles, comprehensive narrative reviews, meta-analytical studies, systematic reviews, clinical studies, and pre-clinical studies, to discuss their findings. We did not include studies on other types of inflammasomes. This narrative review article discusses only the role and therapeutic significance of the NLRP3 inflammasome in various cancers.

## 2. NLRP3 in Melanoma

Melanoma is a melanocyte tumor, and is the fifth leading cancer in the United States. Its incidence rate is rising globally, and 80% of skin cancer deaths are associated with melanoma [18]. Although melanoma incidence has been shown to be dependent on the skin type of melanin-producing cells, recent evidence suggests that mutations due to ultraviolet (UV) radiation (UVA (315–400 nm) and UVB (280–315 nm))/sun exposure are the major cause of neoplasm development [19]. Further, UV beds (indoor tanning stations) could also cause skin cells’ DNA damage and risk of developing melanoma. Specifically, mutations associated with genes such as BRAF, NRAS, KIT, and MAPK have been associated with melanoma [20,21]. Further, the melanoma subtypes and prognosis are dependent on the location of these mutations [21]. Besides genetic mutations, recent studies also indicated the significance of macrophages and other immune cell-mediated inflammatory responses as a major cause of melanoma [22,23]. In some cases, the immune cells could act as biomarkers of melanoma [24]. For example, Reschke et al. [25] have shown that tissue-resident memory T-cells could be potential biomarkers of melanoma.

Macrophage polarization into M1 and M2 phenotypes also plays a significant role in the progression of melanoma [26,27]. The M1 phenotype is pro-inflammatory, while the M2 phenotype is anti-inflammatory and pro-angiogenic, and both cell types can influence tumor progression and immune response. However, in melanoma, increased CD163+ tumor-infiltrating macrophages (M2 phenotype) lead to the loss of Melan-A expression, causing pronounced invasive tumor phenotype [28]. Further, it has been shown that complement component C3a has been identified as playing a role in melanoma progression by affecting immune cell activation and tumor promotion [29]. Similarly, a few studies have also shown that the presence of mature dendritic cells (DCs) and Langerhans cells could act as prognostic indicators [30,31]. A high number of mature DCs correlates with better patient outcomes. Further, it has been shown that pro-inflammatory cytokines and chemokines like IL-1β, IL-6, MCP-1, and TNF-α also play a major role in the inflammation-mediated pathogenesis of melanoma [32]. In addition, NLRP3-mediated immune and inflammatory responses also play an important role in the growth and progression of melanoma, as well as its resistance to immunotherapy. For example, Chow et al. [33] have demonstrated that NLRP3 deficiency prevents methylcholanthrene-induced sarcoma and metastasis in a mouse model by natural killer cell-mediated regulation of immune responses. Similarly, Drexler et al. [34]. have shown that IL-1r and caspase-1 deficient mice are partially protected from epithelial skin cancer. Further, inhibition of NLRP3 inflammasome by thymoquinone prevents mouse and human melanoma cell migration. The NLRP3 inflammasome has been associated with both positive and negative predictive outcomes in melanoma patients. For example, elevated NLRP3 gene-expression scores have been linked to a favorable prognosis [35]. However, other studies have reported that NLRP3 inflammasome activation and the production of IL-1β and IL-18 can promote melanoma growth [36,37]. Further, NLRP3 and IL-1β have been shown to be overexpressed in melanoma samples, and targeting tumor-associated NLRP3 prevents melanoma growth by reducing the expansion of myeloid-derived suppressor cells (MDSCs) [38].

The NLRP3 inflammasome has also been shown to establish a pulmonary metastatic niche and drive disease hyper-progression in response to immunotherapy [39]. Okamoto et al. [40] have also been implicated in promoting late-stage melanoma progression through the cleavage of caspase-1 and active IL-1β. On the other hand, Manica et al. [41] have shown that curcumin prevents cutaneous melanoma by downregulation of NLRP3 activation and decreased IL-1β and IL-18 secretion. Similarly, rosmarinic acid has been shown to prevent the growth and migration of human metastatic melanoma cells by preventing the NLRP3 expression [42]. Moreover, combining anti-programmed cell death-(PD)-1 therapy with NLRP3 inhibition suppressed primary melanoma progression and distant melanoma metastases more effectively than anti-PD-1 monotherapy [38].

Thus, current findings suggest a dual role for NLRP3 in melanoma, where its activation can promote or inhibit tumor growth and progression, depending on the context (Figure 2). In the context of immunotherapy, the tumor-intrinsic NLRP3-heatshock protein 70 (HSP70) toll-like receptor 4 (TLR4) axis has been associated with disease hyper-progression and survival in patients undergoing anti-PD-1 immunotherapy [39]. For example, increased NLRP3 activity in melanoma due to immunotherapy has been shown to regulate the expressions of PD-L1 and Cd155 in the tumor microenvironment [43,44]. Thus, these studies indicate that NLRP3 is involved in the immunotherapy resistance to melanoma. However, additional pre-clinical and clinical studies are required to clearly understand the significant role of NLRP3 in melanoma progression, as well as providing resistance to immunotherapy.

## 3. NLRP3 in Leukemia

Leukemia is a blood-related cancer characterized by the neoplastic transformation of hematopoietic progenitor cells and their infiltration into the bone marrow [45]. According to the National Cancer Institute, as of 2024, leukemia continues to be a significant health concern, with varying incidence and mortality rates for different types of disease. In the United States, it is estimated that there will be 62,770 new cases of leukemia and approximately 23,670 deaths from the disease (https://seer.cancer.gov/statfacts/html/leuks.html (accessed on 22 August 2024)). The most common subtypes of leukemia are acute myeloid leukemia (AML) and chronic lymphocytic leukemia (CLL), with CLL accounting for more mortality. Leukemia can affect people of all ages but is most common in children and older adults. When compared to females, males generally have higher incidence rates. Acute lymphoblastic leukemia (ALL) is more prevalent in children, while AML is more common in adults. CLL and chronic myeloid leukemia (CML) could primarily affect older individuals [45]. Moreover, inherited mutations, environmental factors, age, and unhealthy lifestyle choices are the major risk factors for the development of leukemia.

Further, the immune and inflammatory responses could also play a significant role in leukemia progression, by providing immune surveillance [46,47]. However, leukemia cells can also evade immune detection and chronic inflammation and create a supportive environment for leukemia growth [47]. Recent studies have shown that advanced treatment strategies such as targeted therapies and immunotherapies have improved survival rates, particularly for childhood leukemia [48,49,50]. Nevertheless, significant geographical disparities in incidence and outcomes highlight the necessity of early diagnosis and treatment.

Further, recent studies also indicate that NLRP3 inflammasome activation has been implicated in various types of leukemia, including myelodysplastic syndrome, myeloproliferative neoplasms, AML, CML, and ALL [51,52,53]. Activation of the NLRP3 inflammasome has been associated with promoting proliferation, inhibiting apoptosis, and increasing drug resistance in primary leukemia cells [54]. NLRP3 plays a crucial role in AML by promoting a pro-inflammatory environment through inflammasome activation, regulating cell death via pyroptosis, and aiding immune evasion by leukemic cells. Aberrant NLRP3 activity supports AML progression and survival. Wang et al. [55] have shown that genetic polymorphisms of cytokine genes IL-18 rs1946518 and IL-1β rs16944 are linked with the prognosis and survival of AML patients. Further, Zhong et al. [54] have demonstrated the significance of NLRP3 inflammasome-activated IL-1β in the AML progression. They have shown that the knockdown of NLRP3 prevents leukemia burden in an AML mouse model. This study suggested that targeting NLRP3 is beneficial in controlling AML. Liu et al. [56] have shown that NLRP3-activated bone marrow-derived cells promote Th1 cell differentiation via IL-1β secretion. This, in turn, increases IFN-γ production, induces apoptosis, and inhibits the proliferation of leukemia cells, suggesting a potential approach for AML immunotherapy. Similarly, Liu et al. [57] have demonstrated that stress worsens AML by increasing leukemic cell infiltration and reducing cell survival in a mouse model of chronic restraint stress. This is accompanied by increased NLRP3 inflammasome activation and IL-1β levels. Further, they have shown that knockdown of high-mobility group box 1 (HMGB1) protein, which is highly expressed in newly diagnosed AML patients, suppressed NLRP3 inflammasome and IL-1β expressions, which mitigated the adverse effects of stress on AML progression.

Additionally, Hamarsheh et al. [58] have demonstrated that the NLRP3 inflammasome is implicated in other hematological diseases, such as chronic myelomonocytic leukemia and juvenile myelomonocytic leukemia, particularly through the Kirsten rat sarcoma viral oncogene homologue (KRAS)/RAC1/reactive oxygen species (ROS)/NLRP3/IL-1β axis. This pathway is involved in myeloproliferation and cytopenia, and its effects are reversible in models with NLRP3 deficiency. This study also indicates that the KRAS/ROS/NLRP3/IL-1β axis is also active in AML cells, and suggests that KRAS activation leads to RCA1 activation, which, via NADPH oxidase activation and ROS generation, promote NLRP3-mediated release of innate immune cytokines. In addition, Jia et al. [59] have shown that newly diagnosed AML patients exhibited higher expression of NLRP3 inflammasome and aryl hydrocarbon receptor (AHR) in the bone marrow and peripheral blood, along with elevated IL-18 levels and an increased Th22/Th1 ratio compared to patients in complete remission and normal controls. These studies suggest that NLRP3, via AHR, could play an important role in T-helper cell differentiation and AML progression. Additionally, the NLRP3 inflammasome has been linked to the development of myeloid leukemias, such as chronic myelomonocytic leukemia (CMML), juvenile myelomonocytic leukemia (JMML), and AML, particularly in the context of KRAS mutations [60,61,62].

Glucocorticoids are one of the options for treating patients with ALL, and the treatment is based on the sensitivity and resistance of ALL cells to glucocorticoids [63]. Furthermore, NLRP3 inflammasome and caspase-1 activation have been associated with glucocorticoid resistance in leukemia cells, which has implications for treatment response and disease recurrence [64]. Interestingly, caspase-1 activation could cause glucocorticoid receptor cleavage, which leads to reduced glucocorticoid-induced transcriptional activation and enhanced glucocorticoid resistance. On the other hand, caspase-1 inhibition has been shown to restore glucocorticoid sensitivity by increasing the expression of receptors [65]. Current findings on ALL indicate that glucocorticoid resistance, linked to poorer prognosis, correlates with elevated caspsase-1 and NLRP3 expression, due to reduced methylation of their promoters in leukemia cells [64,65]. In the context of chemotherapy, the NLRP3 inflammasome has been implicated in contributing to therapy resistance in ALL. For example, Hu et al. [66] have suggested that in patients with ALL, post-chemotherapy IL-18 levels in plasma were significantly increased when compared to pre-chemotherapy levels. Chemotherapy induces upregulation of NLRP3, and interestingly, the triterpenoid analog CDDO-2P-Im (1-(2-cyano-3,12-dioxooleana-1,9(11)-dien-28-oyl)-4(-pyridin-2-yl)-1H-imidazole) inhibits NLRP3 inflammasome activation, reduces ALL cell growth in vitro, and promotes apoptosis in steroid-sensitive and steroid-resistant leukemia cells [67]. Thus, this study indicates that inhibition of NLRP3 inflammasome could reverse glucocorticoid resistance and, therefore, could improve the treatment options for ALL. In addition, genetic polymorphisms of NLRP3 inflammasome genes have been investigated for their association with pediatric ALL [68]. These studies provided new insights into the genetic basis of leukemia susceptibility and outcomes. Similarly, genetic polymorphisms associated with NLRP3 have been linked to CML [69].

Very few studies have suggested that there is also a role for NLRP3 in CLL progression. Adinolfi et al. [70] have investigated P2X purinoceptor 7 (P2X7R) expression and its function in 21 patients with B-cell CLL. They found that lymphocytes from those with the evolutive variant had higher resting cytoplasmic Ca++ concentration and more significant Ca++ influx upon P2X7R stimulation, correlating with increased P2X7R expression. Further, they have shown that extracellular ATP reduced the spontaneous proliferation of lymphocytes in patients with the evolutive variant, but did not affect those with the indolent form. This study suggests an association between P2X7R expression and function in the progression of CLL. Similarly, Salaro et al. [71] have also shown overexpression of P2X7R and inhibition of NLRP3 in the lymphocytes of 23 CLL patients. Thus, recent studies suggest that the NLRP3 inflammasome-mediated immune and inflammatory responses play a significant role in leukemia, contributing to disease progression, drug resistance, and therapy response (Figure 3). These studies indicated that, depending upon the type of leukemia and resistance to therapy, NLRP3 inflammasome could be targeted as a potential therapeutic approach.

## 4. NLRP3 in Breast Cancer

Breast cancer is the most diagnosed cancer globally. The American Cancer Society estimated that in 2024, approximately 310,720 new cases of invasive breast cancer will be diagnosed in women in the United States, along with 56,500 new cases of non-invasive breast cancer. Further, breast cancer is still the most common cancer in American women, accounting for approximately 30% of all new female cancer diagnoses each year. While the five-year relative survival rate for localized breast cancer is 99%, the overall number of breast cancer-related deaths in 2024 is projected to be 42,250 for women and 530 for men (https://www.cancer.org/cancer/types/breast-cancer (accessed on 30 August 2024)). Further, inflammatory breast cancer is a lethal form of breast cancer rarely seen in young women under age 40. Pathological inflammation, edema, and breast redness are associated with it. It can progress rapidly and metastasize in various organs.

The NLRP3 inflammasome-mediated inflammatory response plays a significant role in breast cancer progression and metastasis. Recent studies have shown the significance of NLRP3 inflammasome and its involvement in breast cancer progression and metastasis, indicating its potential as an innovative therapeutic target [72,73,74]. NLRP3 inflammasome activation has been associated with promoting breast tumor growth, progression, and aggressiveness, as well as with the pyroptotic death of cancer cells [72,73,74]. Additionally, inhibiting the NLRP3 inflammasome has been shown to suppress breast cancer cell growth and enhance anticancer immunity [75,76]. Innate immune cytokines such as IL-1β and IL-18 have been shown to promote breast cancer progression in animal models [77,78]. They have also been implicated in promoting the migration and invasion of breast cancer cells and induction of IL-1β secretion promote these cells’ epithelial–mesenchymal transition (EMT) [79]. Further, Reed et al. [80] have shown that inhibition of IL-1β prevents iFGFR1-induced early-stage mammary lesions. Zhou et al. [81] have shown that IL-1β is responsible for the increased invasiveness of breast cancer cells, leading to bone metastasis. Zerumbone is an orally active natural cyclic sesquiterpene isolated from *Zingiber zerumbet*. It exerts anti-proliferative, anti-inflammatory, anti-cancer, anti-bacterial and anti-mutagenic activities. Jeon et al. [82] have demonstrated that zerumbone prevents IL-1β-induced triple-negative breast cancer cell invasion. On the other hand, zerumbone has been shown to prevent IL-1β-induced invasion of triple-negative breast cancer cells by inhibiting the expression of IL-8 and matrix metallopeptidase 3 (MMP3) [83]. Further, IL-1β has been shown to promote tumor invasiveness in breast cancer by activating the IL-1β/IL-1RI/β-catenin signaling pathway, leading to β-catenin accumulation and nuclear translocation [84]. Garcia-Morales et al. [85] have also shown that cannabidiol (CBD) prevents breast cancer cell viability, blocks the IL-1β/IL-1RI/β-catenin signaling pathway, restores epithelial organization, and prevents nuclear translocation of β-catenin, ultimately indicating a potential therapeutic role for CBD in counteracting the malignant features induced by IL-1β. Another study by Pham et al. [86] has shown the role of adiponectin, particularly its globular form (gAcrp), in suppressing breast cancer cell growth by inhibiting inflammasome activation. The study demonstrates that gAcrp significantly inhibits the activation of NLRP3 inflammasome components. This inhibition blunts the inflammatory response, leading to decreased cell viability, increased apoptosis, and causing cell cycle arrest via sestrin2 (SESN2) induction and adenosine monophosphate-activated protein kinase (AMPK) activation-dependent mechanisms. Similarly, Chen et al. [87] have shown that breast cancer susceptibility gene BRCA1 deficiency disrupts mitochondrial dynamics, leading to impaired stress-induced mitophagy and the activation of the NLRP3 inflammasome, which promotes a tumor-associated microenvironment and facilitates cancer progression. This study, thus, suggests that inhibiting inflammasome activation could be a potential therapeutic strategy to control tumor recurrence and metastasis in BRCA1-related breast cancer.

Further, NLRP3 inflammasome inactivation driven by microRNA (miR)-223-3p has been shown to reduce tumor growth and increase anticancer immunity in breast cancer [88]. Xia et al. [89] have shown that P2X7 receptor activation promotes breast cancer cell invasion and migration by activating protein kinase B (AKT) signaling. Moreover, blocking P2X7 has demonstrated inhibition of breast cancer growth in mice through the NLRP3/caspase-1 pathway [90,91]. Thus, previous studies have suggested that the NLRP3 inflammasome exerts a significant influence on breast cancer progression, metastasis, and immune response. Modulating NLRP3 inflammasome activity has been proposed as a potential therapeutic strategy for breast cancer treatment (Figure 4). However, the precise mechanisms involved in NLRP3 inflammasome activation and its specific impact on the pathophysiology of breast cancer development need further investigations to demonstrate its potential use as a therapeutic target.

## 5. NLRP3 in Lung Cancer

Worldwide, more than 1 million people die due to lung cancer each year. It is estimated that cigarette smoking is the main reason for almost 90% of lung cancer risk in men and 70 to 80% in women [92]. Lung cancer is the second most common cancer and the cause of the most cancer-related deaths in both men and women in the U.S. [93]. The American Cancer Society estimated that in 2024, ~234,580 new cases and ~125,070 deaths will occur from lung cancer, non-small-cell and small-cell combined, in the U.S. The transition of normal lung epithelium from adenoma to carcinoma is associated with a variety of molecular and biochemical events, such as genetic alterations, epithelial cell proliferation/differentiation, and inflammation [94,95]. Primary initiators of carcinogenesis include (a) cells that suffered irreparable DNA damage due to increased free radicals, which cause activation of specific nucleases and damage DNA, RNA, proteins, and lipids; (b) loss of extracellular stimulation that regulates cell growth; (c) upregulation of growth factors and their receptors, and (d) autosomal dominant inheritance of cancer genes in multiple family members [94,95]. Dietary and environmental factors also play an essential role in predisposition to carcinogenesis. Furthermore, chronic inflammatory diseases such as chronic obstructive pulmonary disease (COPD), and infectious diseases such as tuberculosis are associated with an elevated risk of lung cancer [96,97,98,99]. Although it is not clear how cancer is initiated in the setting of chronic inflammation, increasing evidence strongly supports the association between lung cancer and inflammation [100,101].

Inflammation-mediated lung cancer is mainly associated with lung epithelial cells and alveolar macrophages. Several studies have shown NLRP3 inflammasome activation in these cells upon oxidant exposure. For example, cigarette smoke chemicals, including nicotine, have been shown to induce NLRP3 activation in human alveolar and bronchial epithelial cells [102,103,104]. Cigarette smoke-induced increase in oxidative stress and mitochondrial dysfunction could be responsible for the increase in NLRP3 activation, leading to lung carcinogenesis [105]. Further, increased expression of NLRP3 has been seen in mouse models of COPD [106]. Similarly, Rao et al. [107] have shown that in a mouse model of radiation-induced lung injury, there is a significant increase in the activation of NLRP3 inflammasome. Li et al. [108] have also shown that targeting NLRP3 inflammasome could enhance potential radiotherapy in non-small-cell lung cancer.

On the other hand, Zhang et al. [109] have indicated that cigarette smoke prevents the inflammasome complex formation and release of IL-1β in mice alveolar macrophages. Additionally, long-term cigarette smoke has been shown to cause NLRP3 activation in the oral mucosal epithelium [110]. Since oxidative stress and chronic inflammation are common in both COPD and lung cancer, the risk of developing lung cancer in COPD is always higher. Although epigenetic alterations and genetic mutations are major causes of lung cancer, COPD patients have an increased risk of developing lung cancer. However, several studies have supported the role of NLRP3 in lung cancer. A study by Wang et al. [111] has shown that increased NLRP3 activation is responsible for the growth and migration of lung cancer cells.

Another study by Dutkowska et al. [112] has shown increased expression of IL-1β in non-small-cell lung cancer tumor tissues. Similarly, Kong et al. [113] have shown increased NLRP3 inflammasome in lung adenocarcinoma and small-cell lung cancer. The significance of NLRP3 in lung cancer is further confirmed by a mouse study where NLRP3-null mice inhibit lung cancer growth induced by benzo(a)pyrene without or with lipopolysaccharide (LPS) cotreatment [114]. Further, inflammasome-released cytokine IL-1β has been shown to promote metastasis of lung cancer cells through adhesion, invasion, and angiogenesis processes [115]. Similarly, Li et al. [116] have shown that NLRP3-mediated release of IL-1β and IL-18 increases the proliferation and migration of non-small-cell lung cancer cells. Recently, Metwally et al. have identified the fact that IL-1β rs16944 variants are associated with the development of non-small-cell lung cancer [117]. Furthermore, NLRP3 activation has been associated with promoting the proliferation and migration of lung cancer cells, impacting cancer cell viability and survival [118]. Conversely, some studies suggest a protective role of NLRP3 in lung cancer. NLRP3 expression is downregulated in non-small-cell lung cancers, with its expression positively correlating with overall survival [119]. Several studies indicate that lung cancer chemopreventive effects of some of the antioxidants are mediated by inhibiting NLRP3 inflammasome. For example, the anticancer effects of plant products reniformin A, saikosponin-D, and EEBR are exerted by inducing NLRP3-mediated pyroptosis in non-small-cell lung cancer cells [120,121,122]. Similarly, Zhao et al. [123] have shown that by inhibiting NLRP3, melatonin could regulate angiogenesis and lymphangiogenesis in lung adenocarcinoma. The role of NLRP3 in lung cancer is not clearly known; further studies are needed to elucidate the mechanisms underlying NLRP3 inflammasome’s beneficial and harmful impacts on lung cancer pathophysiology (Figure 5).

## 6. NLRP3 in Colon Cancer

Colon cancer is the third most common cancer and the second leading cause of cancer deaths in the world [124]. According to the National Cancer Institute, it is estimated that around 152,810 new cases of colon cancer and 53,010 deaths could happen due to this cancer (https://seer.cancer.gov/statfacts/html/colorect.html (accessed on 9 May 2024)). The most common cause of colon cancer is due to genetic predispositions, such as familial adenomatous polyposis and hereditary non-polyposis colon cancer (Lynch syndrome) [125,126]. Further, inflammatory bowel diseases (IBDs) such as ulcerative colitis and Crohn’s disease, poor lifestyle, and increased consumption of processed meat and alcohol are some of the major risk factors for developing colon cancer [127]. The risk of colon cancer has been shown to increase with age. Individuals fifty years and older have been shown to have an increased risk of colon cancer [124]. Increased oxidative stress and inflammatory response have also been associated with the elevated risk of colon cancer [128,129]. IBD is an inflammatory complication, and most of the cases of colon cancer are also associated with it. Further, recent studies using various antioxidants have also suggested that oxidative stress-regulated and NLRP3 inflammasome-mediated innate immune and inflammatory response could also contribute to colon cancer growth and spread [130].

In IBD, the activation of NLRP3 inflammasome-mediated IL-1β release has been shown to be a major risk factor for developing colon cancer [131]. NLRP3 inflammasome has a potential impact on colon carcinogenesis, inflammation, and possible therapeutic interventions, including chemo- and immuno-therapies. Further, in IBD patients, NLRP3 activation has been shown in the intestinal epithelial cells, and polymorphisms in IL-1β and 1L-18 have been associated with ulcerative colitis patients. Indeed, increased IL-1β has been demonstrated in patients with IBD [132]. Further, IL-1β- and NLRP3-null mice have shown significantly reduced inflammatory response and colitis development [133,134]. Administration of IL-1β to NLRP3-null mice has shown protection against oxazolone-induced colitis [135]. Further, small molecular inhibitors such as andrographolide have been shown to protect against azoxymethane and dextran sodium sulfate-induced colon cancer by inhibiting the NLRP3 inflammasome-mediated inflammatory response [136]. Further, several plant-based antioxidants and synthetic small molecular inhibitors have been shown to prevent IBD by inhibiting the NLRP3 inflammasome and expression of inflammatory markers such as IL-1β, IL-6, INF-γ, and IL-2 [130,137]. Hu et al. [138] have also shown that drugs that prevent the palmitoylation of NLRP3 could reduce the symptoms associated with dextran sodium sulfate-induced colitis in a mouse model. Further, in an acetic acid-induced colitis model and caco-2 colon epithelial cells, cyclooxygenase-2 (COX-2) inhibitor (mangiferin) prevents activation of NLRP3 and expression of inflammatory markers such as IL-1β, TNF-α, IL-16, and IFN-γ [139]. Similarly, quercetin has been shown to prevent NLRP3 inflammasome in HCT116 and HT29 colon cancer cells and a nude-mouse xenograft model [140].

Further, Allen et al. have shown that NLRP3 inflammasome has been suggested to function as a negative regulator of tumorigenesis during colitis-associated cancer (CAC) [141]. Similarly, activation of inflammasome-mediated caspase-1 by promoting cancer cell apoptosis could prevent colon cancer [142]. Knockdown or inhibition of NLRP3 activation has been shown to increase colonic inflammation, indicating a potential therapeutic avenue for managing colitis-associated cancer [143]. Similarly, it has been shown that Zinc finger protein 70 increases the IL-1β secretion in macrophages, promoting colon cancer cell proliferation (HCT-116). They have also demonstrated that the knockdown of zinc finger protein 70 could inhibit NLRP3 and azoxymethane/dextran sulfate sodium (AOM/DSS)-induced colon cancer [144]. In addition, gut microbiota plays an essential role in controlling inflammation. A few studies also indicate that NLRP3 inhibition prevents IBD by regulating the gut microbiota [145]. Similarly, Gao et al. [146], using a DSS-induced colitis model, have also demonstrated that by improving the gut microbiota, an herbal Chinese medicine could prevent NLRP3-mediated expression of inflammatory markers. They have also shown that NLRP3 is also expressed in both immune cells and colonic epithelial cells, suggesting a possible association with colon cancer. The NLRP3 inflammasome has been associated with the regulation of intestinal homeostasis, with its activation linked to inflammation-induced cancer [147].

Furthermore, the NLRP3 inflammasome has been implicated in the proliferation, migration, and invasion of cancer cells in gastrointestinal cancers, including colon cancer [148]. NLRP3 inhibition has also been shown to prevent the growth and migration of non-small-cell lung cancer cells, indicating a broader role in promoting cancer progression [149]. Additionally, Zhang et al. have shown that NLRP3 inflammasome is involved in the spread of colon cancer by increasing the epithelial–mesenchymal transition (EMT), a process essential for migration and invasion [150]. They have shown that NLRP3 activation and release of IL-1β mediates macrophage-mediated increase in the invasion and migration of colon cancer cells through regulating the EMT (Figure 6). Similarly, Deng et al. [151] have also shown that NLRP3 activation in macrophages mediates colon cancer migration and invasion. So far, these studies suggest that NLRP3 inflammasome mediates colon cancer migration and invasion.

## 7. NLRP3 in Gastric and Pancreatic Cancers

Gastric cancer is the fifth most common cancer worldwide [152]. Most gastric cancers are associated with the *Helicobactor pylori* (*H. pylori*) infections [152,153]. Moreover, genetic mutations in the CDH1 gene (familial hereditary diffuse gastric cancer), alcohol consumption, processed meats, and high salt preserved foods could also contribute to the risk of developing gastric cancer [154]. Inflammation is a key in the development of gastric cancer, especially during *H. pylori* infections. The bacteria are known to colonize the lining of the stomach and induce prolonged inflammation, leading to chronic gastritis. Thus, chronic inflammation could lead to damage to the gastric epithelial cells, causing intestinal metaplasia and gastric adenocarcinoma [155]. NLRP3-mediated immune response has also been associated with chronic gastritis. Specifically, a few studies have shown that *H. pylori* activates NLRP3 and releases inflammatory IL-1β and IL-18 cytokines [156,157].

Further, several studies have shown that NLRP3 is highly expressed in stomach adenocarcinoma, and could serve as a potential prognostic biomarker for gastric cancer [148,158,159]. *H. pylori* infection has been associated with enhanced ROS production and NLRP3 expression in gastric cancer, triggering alterations in the M1 and M2 macrophage polarization [160]. Silencing NLRP3 has been demonstrated to reduce the effects of CagA on gastric cancer-cell migration and invasion, suggesting a potential role of NLRP3 in promoting the metastatic potential of gastric cancer [161]. Additionally, the NLRP3 inflammasome has been linked to the regulation of IL-1β production, which is implicated in gastric cancer development [162,163]. IL-18 has also been shown to contribute to increased inflammatory response in a DSS-induced IBD model [164]. Further, Chen et al. [165] have also shown that *H. pylori* causes mitochondrial oxidative damage and dysfunction, helping bacteria evade the host’s immune response. They have shown that by increasing mitophagy, *H. pylori* prevents NLRP3 inflammasome, leading to the development of gastric cancer. Similarly, Qi et al. [166] have shown that flavonoids such as 3.4.5.7-tetrahydroxyflavone induce mitochondrial damage, which leads to activation of NF-κB/NLRP3-induced pyroptosis in human gastric adenocarcinoma cells. Liu et al. [167] also indicate that Baicalin promotes gastric cell pyroptosis by activating NF-κB and NLRP3 signaling pathways. Further, bile acids have been shown to cause mitochondrial tethering to mitochondria, which can cause increased calcium overload and NLRP3 activation [168,169].

Further, paclitaxel has been shown to prevent NLRP3-mediated migration and invasion of gastric cells, suggesting NLRP3’s involvement in the metastasis of gastric cancer [170]. Furthermore, overexpressed long non-coding RNA ADAMTS9-AS2 has been shown to inhibit gastric cancer development by regulating the miR-223-3p/NLRP3 axis, suggesting a potential tumor-suppressive role for NLRP3 in gastric cancer [171]. A few studies have also indicated that gut microbiota has been shown to be involved in gastric cancer progression (Table 1). For example, *H. pylori* infection has been shown to disrupt gut microbial homeostasis, leading to increased NLRP3-mediated inflammatory response [172,173]. Thus, these studies suggest that NLRP3 plays a critical role in *H. pylori*-induced gastric cancer.

## 8. NLRP3 in Pancreatic Cancer

Pancreatic cancer, or pancreatic ductal carcinoma, is rare, and accounts for 3% of all cancers. According to the American Cancer Society in the U.S., in 2024, an estimated 66,440 people may be diagnosed with pancreatic cancer, and around 51,750 people may die from it (https://www.cancer.org/cancer/types/pancreatic-cancer/about/key-statistics.html (accessed on 9 December 2024)). Although genetic mutations, obesity, and smoking are major risk factors for the development of pancreatic cancer, inflammation of the pancreas, pancreatitis, is one of the major causes [174].

Oxidative stress, as well as inflammatory responses mediated by the NF-κB and NLRP3 pathways, have been shown to induce pancreatitis [175,176,177]. Specifically, pro-inflammatory cytokines expressed by NF-κB, such as TNF-α, IL-6, IFN-γ, and IL-8, and innate immune cytokines released by NLRP3, such as IL-1β and IL-18, play a significant role in pancreatitis. Xu et al. [178] have shown that in pancreatic acinar cells, IL-1β is involved in the formation of trypsin and inhibition of cell growth. Further, Caronni et al. [179] have suggested the importance of tumor-associated macrophage-mediated inflammatory responses in pancreatic cancer. They have demonstrated that pancreatic cancer progression can be inhibited by preventing the interactions between prostaglandin E2 (PGE2) and IL-1β in the tumor-associated macrophages. Previously, a study by Chen et al. [180] has also shown that pancreatic ductal adenocarcinoma cell debris activates NF-κB signaling in the M2 macrophages and releases IL-1β. They have also demonstrated that IL-1β is responsible for EMT and metastasis of pancreatic cancer. Further, Wu et al. [181] have also demonstrated that HMGB1 mediates the NLRP3-mediated processing of IL-1β in macrophages and pancreatic cells, leading to pancreatic injury.

In addition, several studies have noted the influence of NLRP3 in pancreatic cancer (Table 1). For instance, NLRP3 has been associated with increased frequency of the rs35829419-NLRP3 polymorphism in patients with pancreatic cancer, suggesting a potential genetic association with the disease [182]. Additionally, upregulation of NLRP3 signaling has been linked to promoting platelet activation and aggregation, leading to tumor growth and metastasis in a mouse model of pancreatic cancer [183]. These studies suggest that NLRP3 inflammasome has been associated with an increased risk of inflammation-induced pancreatic cancer.

In addition, some studies have suggested the protective role of NLRP3 in pancreatic cancer. For example, downregulation of NLRP3 has been shown to inhibit the proliferation and migration of pancreatic cancer cells, suggesting a potential tumor-suppressive role [184]. Moreover, the NLRP3 inflammasome has been implicated in promoting pancreatic islet damage, indicating its possible involvement in the pathophysiology of pancreatic cancer [185]. The studies on the prevention of pancreatitis further support the role of NLRP3 by using various antioxidants in animal models. For example, Ren et al. [186] have shown that the natural plant product Danshensu prevents the activation of NF-κB and NLRP3 in an acute-pancreatitis mouse model. Similarly, pinocembrin, pramipexole, proanthocyanidins, and Emodin inhibit NLRP3-mediated caspase-1-1 activation and IL-1β production in animal models of acute pancreatitis [187,188,189,190]. These studies indicate the beneficial and harmful effects of NLRP3 in the development of pancreatic cancer. However, additional studies are still needed to clarify the role of NLRP3-mediated immune and inflammatory responses in pancreatic cancer progression and therapy.

## 9. NLRP3 in Prostate Cancer

Prostate cancer is one of the most common cancers in men, and it is estimated that in 2024, around 299,010 new cases of prostate cancer and 35,250 deaths might happen from it (https://www.cancer.org/cancer/types/prostate-cancer/about/key-statistics.html (accessed on 9 December 2024). Recent studies also suggest that systemic inflammation also plays a major role in prostate tumor progression and metastasis [191,192]. When compared to other cancers, the role of NLRP3 in prostate cancer is not well investigated. Although few studies indicate the beneficial role of NLRP3 inflammasome inhibition in prostate cancer using animal models, clinical studies are required to understand its role in the progression of prostate cancer. Expression of inflammasome proteins has been observed in the human prostate epithelial cells [193]. Further, expression of NLRP3 in prostate cancer cells (PC-3) has been shown to be increased under hypoxia [194]. Xu et al. [195] have also shown the expression of NLRP3 in LNCaP and PC3 cells and prostate cancer tissues. Further, they have shown that activation of NLRP3 mediates prostate cancer cell proliferation and migration, and inhibition of NLRP3 prevents it. Similarly, pyroptosis mediated by caspase-1 has been shown to be involved in prostate cancer progression [196]. Moreover, the NLRP3 inflammasome has been implicated in promoting prostate islet damage, indicating its potential involvement in the pathophysiology of prostate cancer [197]. The involvement of NLRP3 in prostate cancer is further supported by a study from Zeng et al. [198]. They have shown that Carvedilol (CVL), a β-adrenergic receptor antagonist, causes NLRP3 inflammasome-mediated pyroptosis in prostate cancer.

Furthermore, the upregulation of NLRP3 signaling has been linked to the development of prostate inflammatory lesions in rat models of pancreatitis when treated with a combination of estrogen benzoate and a high-fat diet [199]. Chronic pancreatic atrophy and inflammation are associated with the development of prostate cancer. Similarly, NLRP3 has been associated with tumor–node–metastasis staging in prostate cancer [200]. Another study by Chaudagar et al. [201] suggested that NLRP3 is highly expressed in tumor-associated macrophages in patients undergoing androgen-deprivation treatment. Here, androgen receptor inhibition has been shown to increase NLRP3 expression, but not activity. Further, they have shown that a combination of androgen-deprivation treatment. along with inhibition of NLRP3. prevents prostate cancer in a mouse model. Similarly, NLRP3 acetylation and inhibition of inflammasome complex formation by androgen receptor inhibitor could block prostate cancer progression in nude-mice xenografts [202]. A recent study by Liu et al. [203] has shown that ulinastatin, a protease inhibitor, prevents prostate cancer in a Rho/Rock/NLRP3 inflammasome pathway in PC-3 cells. In addition, LPS + Nigericin-induced NLRP3 inflammasome has been shown to be inhibited by doxycycline [204]. Similarly, Natriuretic peptides such as Atrial Natriuretic Peptides (ANPs) and B-Type Natriuretic Peptide (BNP) have been found to inhibit NLRP3 inflammasome and its associated inflammatory response in prostate cancer [205]. Thus, a few studies indicate that activation of the NLRP3 inflammasome has been associated with promoting inflammation-induced carcinogenesis in prostate cancer. However, additional studies are required to understand the molecular pathways through which inflammasome-mediated inflammatory response is involved under normal, hypoxia, androgen receptor-mediated progression and metastasis of prostate cancer.

## 10. NLRP3 in Gynecological Cancers

Gynecological cancers, including ovarian, cervical, and endometrial cancers, are significant health concerns, globally [206]. Uncontrolled growth of endometrial tissue outside of the uterine cavity causes endometriosis. Inflammation plays a major influence in the progression of endometriosis [207]. Specifically, alterations in the immune cell function, such as NK cell and T-cell functions, and the phenotype could lead to endometriosis [208]. Few studies have indicated that NLRP3 activation causes endometrial cell apoptosis. Further, NLRP3 activation has been associated with endometriosis, where defective macrophages and endometrial stromal cells elevate the expression of NLPR3 [209]. This study has demonstrated that IL-1β promotes endometrial cell migration and lesion development in a mouse model, and the inhibition of NLRP3 prevents lesion size and cell migration. Estrogen has been shown to activate NLRP3 inflammasome through the estrogen response element, and is involved in the pathology of endometriosis [210]. Further, Zhang et al. [211] have shown that in endometrial tissues, increased angiogenesis is associated with increased Notch1 signaling and NLRP3-induced pyroptosis. Similarly, Li et al. [212]. have shown that NLRP3 inflammasome activation through IncRNA NEAT/miR-141-3p/HTRA1 pathway leads to endometriosis. In addition, Hang et al. [213] have also shown that tripartite motif-containing 24 (TRIM24) is involved in the endometriosis progression through ubiquitination of NLRP3 and pyroptosis.

Further, a cross-sectional study by Fonseca et al. [214] has investigated NLRP3 inflammasome in follicular fluid and granulosa cells in women undergoing controlled ovarian stimulation. They found that IL-1β and IL-18 levels were elevated in the follicular fluid in the endometriosis group, compared to the non-endometriosis group. Similarly, Huang et al. [215] have found that the expression of PGE2 and NLRP3 inflammasome-related proteins, such as cleaved caspase-1, IL-1β, and IL-18, increased in the endometrial tissues. The significance of NLRP3 inflammasome in endometriosis is further confirmed by studies using various antioxidants and NLRP3 inhibitors. For example, fisetin has been shown to prevent oxidative stress, Poly ADP ribose expression, and to cause apoptosis in endometrial cells through the NLRP3 pathway [216]. Similarly, MCC950 (an NLRP3 inhibitor) prevents ovarian endometriosis and the expression of NLRP3 and IL-1β in the endometrial stromal cells and cyst-derived stromal cells. Alpha-lipoic acid has been shown to prevent endometriosis by preventing the NLRP3-mediated release of IL-1β and IL-18 [217].

Ovarian cancer is one of the major causes of cancer-related deaths in women because of late diagnosis and metastasis [218]. Although inflammation has been shown to be a risk factor responsible for developing ovarian cancer, the role of NLRP3 inflammasome in ovarian cancer is not well studied when compared to other types of cancers. However, a few studies have highlighted NLRP3’s role in tumorigenesis and cancer progression in ovarian cancers (Table 2) [219]. Initially, Chang et al. [220] have identified that NLRP3, IL-1β, and IL-18 genes have been upregulated during the transformation of endometriosis to ovarian cancer. Later, Luborsky et al. [221] found that inflammasome complex proteins and related cytokines are elevated in hen and human ovarian cancers. Further, Solini et al. [222] have shown the involvement of adipocyte P2X7R-NLRP3 inflammasome in regulating the chemotaxis and metastasis of epithelial ovarian cancer. It has been shown that increased NLRP3 in ovarian cancer is associated with overall survival [223]. Increased expression of NLRP3 inflammasome and activation of IL-1β has been associated with the progression of ovarian endometriosis, and inhibitor NLRP3 using MCC950 prevents ovarian endometriosis [224]. Similarly, citric acid has been shown to prevent ovarian cancer cell growth through caspase-4, thioredoxin-interacting protein, and NLRP3 inflammasome-mediated pyroptosis [225]. Further, a natural antioxidant, polydatin, has been shown to prevent ovarian and cervical cancers by inhibiting NLRP3 inflammasome activation. The significance of NLRP3 activation in ovarian cancer is further confirmed by a study by Li et al. [226], where they found that NLRP3 is overexpressed in ovarian cancer and is correlated with the poor survival rate and cisplatin resistance to ovarian cancer. This study also demonstrated that silencing of NLRP3 prevents EMT and sensitizes cisplatin chemotherapy. Similarly, another study by Wu et al. [227] has indicated that RAS-associated C3 botulinum toxin substrate 1 (RAC1) expression is associated with paclitaxel resistance to ovarian cancer. They have shown that RAC1 promotes paclitaxel resistance by inhibiting the PAK4/MAPK pathway, as well as caspase-1 and GSDMD-mediated pyroptosis.

Cervical cancer is one of the major cancer-related deaths in females globally. According to the American Cancer Society, in the USA for 2024, it is estimated that around 13,280 new cases of invasive cervical cancer and around 4360 cervical cancer-related deaths could be possible (https://www.cancer.org/cancer/types/cervical-cancer/about/key-statistics.html (accessed on 24 September 2024)). Inflammation is one of the major causes of the progression of cervical cancer. Specifically, viral infections such as human papillomavirus (HPV)-induced inflammation have been associated with the development of cervical cancer. Further, Pontillo et al. [228] have indicated that single-nucleotide polymorphisms in inflammasome genes such as NLRP3 and others could be associated with the HPV-induced progression of cervical cancer. They found that in case-control analysis, IL-1β rs1143643 is linked with HPV protection, while NLRP3 rs10754558 has been linked with a lower risk of infection with HPV. Further, infection with Chlamydia trachomatis has been shown to increase NLRP3-dependent caspase-1 activation in cervical endothelial cells [229]. Similarly, Lu et al. [230] have found that NLRP3 rs10754558 is associated with an increased risk of cervical cancer. The therapeutic benefit of inhibiting NLRP3 inflammasome is further confirmed by a study using the CD200R1 agonist, CD200Fc. This study found that CD200Fc inhibits LPS-induced NLRP3 inflammasome activation and release of IL-1β in SiHa and Caski human cervical cancer cells [231]. Further, Yu et al. [232] have conducted a clinical study using 50 cervical cancer patients and age-matched controls, and they found that miRNA-214 increases the NLRP3 and [232,233] causes pyroptosis, and thus prevents cervical cancer cell proliferation. Further, NLRP3 inflammasome activation has been linked to promoting pyroptosis, a form of programmed cell death, in cervical and ovarian cancer cells, affecting cancer cell viability and survival [232,233]. In addition, NLRP3 has been implicated in promoting epithelial–mesenchymal transition (EMT) in ovarian and cervical cancer cells, a critical process for cancer metastasis and progression [234]. A recent study by Fernandes et al. [235] has indicated that poor overall survival of cervical cancer is associated with overexpression of IL-1β. Further, Ji et al. [236] have demonstrated that the oncogene Foxm1 induces NLRP3 inflammasome activation and promotes CD8+ T-cell mediated immunosuppression in cervical cancer.

**Table 2 ijms-25-13689-t002:** Role of NLRP3 inflammasome in gynecological cancers.

Cancer Type	Role of NLRP3	Citations
Endometriosis	Elevated NLRP3 expression in defective macrophages and endometrial stromal cells is associated with endometriosis.	[209]
	IL-1β promotes endometrial cell migration and lesion development, and NLRP3 inhibition reduces lesion size.	[209]
	Estrogen activates NLRP3 through the estrogen response element, linking it to endometriosis pathology.	[210]
	NLRP3-induced pyroptosis is associated with increased Notch1 signaling and angiogenesis in endometrial tissues.	[211]
	NLRP3 inflammasome activation through IncRNA NEAT/miR-141-3p/HTRA1 pathway contributes to endometriosis.	[212]
	TRIM24, through NLRP3 ubiquitination and pyroptosis, causes endometriosis.	[213]
	Elevated IL-1β and IL-18 levels are seen in follicular fluid of endometriosis patients undergoing ovarian stimulation.	[214]
	PGE2- and NLRP3-related protein expressions are increased in endometrial tissues.	[215]
	Antioxidants like fisetin, MCC950, and alpha-lipoic acid inhibit NLRP3-mediated inflammation and prevent endometriosis.	[217]
Ovarian cancer	NLRP3, IL-1β, and IL-18 are upregulated during transformation of endometriosis to ovarian cancer.	[220]
	Elevated inflammasome complex proteins are seen in hen and human ovarian cancers.	[221]
	Adipocyte P2X7R-NLRP3 inflammasome shown to regulate chemotaxis and metastasis.	[222]
	High NLRP3 expression is linked to poor survival and cisplatin resistance, and silencing NLRP3 sensitizes chemotherapy.	[226]
	Polydatin inhibits NLRP3 inflammasome activation and prevents ovarian cancer progression.	[225]
	Citric acid prevents ovarian cancer cell growth via NLRP3-mediated pyroptosis.	[225]
	RAC1 expression linked to paclitaxel resistance through PAK4/MAPK pathway and NLRP3-mediated pyroptosis.	[227]
Cervical cancer	NLRP3 polymorphisms are linked to HPV infection risk and cervical cancer progression.	[228]
	Chlamydia trachomatis infection increases NLRP3-dependent caspase-1 activation.	[229]
	NLRP3 rs10754558 is associated with increased cervical cancer risk.	[230]
	CD200Fc inhibits NLRP3 activation in cervical cancer cells by reducing IL-1β production.	[231]
	miRNA-214 increases NLRP3, promotes pyroptosis and reduces cervical cancer cell proliferation.	[232]
	NLRP3-mediated EMT is implicated in cancer metastasis.	[234]
	Foxm1 induces NLRP3 activation by promoting immunosuppression.	[236]

## 11. NLRP3 in Head and Neck Cancer

Head and neck cancer, where most of the cases are of squamous-cell carcinoma of the epithelial lining of the oral, pharynx, and larynx, is the seventh most common cancer worldwide [237]. Although tobacco chewing, alcohol, environmental smoke, and viral infections are the major causes of head and neck cancer (HNSCC), recent studies also indicate that inflammation could also play a role [238]. Specifically, the NLRP3 inflammasome activation in squamous-cell carcinoma tissues has been associated with the progression of HNSCC [239]. Some studies indicate that the involvement of IL-1β in cancer progression and the inhibition of IL-1β could prevent it [240]. Chen et al. have shown that expression of NLRP3 is increased in HNSCC tissues while IL-1β is increased in the peripheral blood of the patients [241]. They have also shown that increased NLRP3 inflammasome is associated with HNSCC progression, and its inhibition leads to decreased accumulation of immunosuppressive cells and increased effector T-cells. In a separate study, Chen et al. [242] have also shown that increased NLRP3 in tumor-associated macrophages leads to poor prognosis and increased HNSCC growth. Increased expression of NLRP3 and increased serum IL-1β has been observed in oral squamous-cell carcinoma (OSCC) patients [243]. Thus, the prognostic value of inflammasomes in head and neck carcinoma has also been highlighted, indicating their relevance in predicting patient outcomes [242,243]. Moreover, NLRP3 has been implicated in promoting the proliferation and migration of esophageal squamous-cell carcinoma, suggesting its involvement in cancer progression [244].

Similarly, Feng et al. have shown that increased NLRP3 and IL-1β production is correlated with the 5-fluorouracil chemoresistance in OSCC, and NLRP3- and IL-1β-null mice showed decreased tumor incidence [245]. On the other hand, Yang et al. [246] have shown that bitter melon-derived extracellular vesicles increase 5-FU-induced therapeutic efficacy and reduce resistance to 5-FU in OSCC cells. Further, in OSCC, zymosan from candida albicans fungal cell wall has been shown to promote IL-1β production and proliferation of oral squamous-cell carcinoma cells [247]. Chow et al. [248] have demonstrated that calcium regulator CD38 prevents NLRP3-mediated pyroptosis in HNSCC cell lines. Xiao et al. [249] have also shown that IL-6 is important in OSCC progression by increasing the JAK2, STAT3, Sox4, and NLRP3 signaling pathways. The metastasis of OSCC has been shown to be prevented by BAY-117082, an NLRP3 inflammasome inhibitor, in a mouse orthotopic model [250]. Similarly, melatonin has been shown to prevent OSCC by regulating the MIF, NLRP3, and IL-1β pathway [251]. Antioxidant bacopa monnieri prevents NLRP3 inflammasome through mitophagy and oral cancer growth in a 4-nitroquinolin-1-oxide and arecoline-induced mouse model [252]. Further, Coenzyme Q0 prevents HNSCC growth in xenograft mice by inhibiting NLRP3 expression [253]. These studies suggest that the NLRP3 inflammasome plays a significant role in head and neck cancers, with evidence suggesting its involvement in tumorigenesis and patient prognosis. Further research is needed to fully understand the mechanisms underlying NLRP3 inflammasome activation and its impact on head-and-neck-cancer pathophysiology.

## 12. Conclusions and Future Perspectives

Recent studies indicate that NLRP3 inflammasome is critically involved in the various aspects of cancer progression and metastasis by either altering the immune response at the tumor microenvironment or inducing the proliferation or pyroptosis of cancer cells. Although some studies clearly indicate that inhibition of NLRP3 is an innovative approach to control cancer growth and spread, its potential as a therapeutic target and a prognostic marker across different cancers needs further investigation. Those studies will help to understand the significance of NLRP3-mediated immune and inflammatory responses in cancer progression, and also help to identify the significance of inflammasomes in advancing cancer immunotherapy.

The activation of the NLRP3 inflammasome plays a pivotal role in cancer progression and the immune response, in a dual way. In one way, NLRP3 activation, by releasing pro-inflammatory cytokines such as IL-1β and IL-18 could lead to chronic inflammation, promoting tumor growth and spread. In another way, it can boost anti-tumor immunity by promoting cell death and attracting immune cells to the tumor site. This binary function is of utmost importance for the potential development of NLRP3 inhibitors as part of cancer therapy, which could inhibit inflammation-linked tumor progression. However, not all cancer types respond to similar immunological responses. Therefore, additional studies are required to identify which specific forms of cancer could benefit from NLRP3 inhibitors. Understanding treatment approaches could reduce the chemo-, hormone- and immune-resistance of cancer cells, enhance therapeutic efficacy, and diminish the possible side effects. Further, developing targeted NLRP3 inhibitors and their combination with other treatments could offer better treatment options for advanced cancer treatments with improved patient outcomes.

Although the prospects of using NLRP3 inhibitors in cancer therapy are promising, special attention needs to be taken when treating different cancers, as inflammasome could play a dual role in both promoting and inhibiting cancer. In some cancers, where inflammation plays a significant role, NLRP3 inhibitors could be crucial by diminishing inflammation and slowing tumor growth and spread. In these circumstances, NLRP3 inhibitors could reduce the pro-inflammatory cytokines and their mediated inflammatory responses that promote tumor growth and metastatic spread. However, additional clinical studies are necessary to address the significance of NLRP3 inhibitors in this direction.

In more resistant and advanced cancers, the combination of NLRP3 inhibitors, chemo drugs, and immune checkpoint inhibitors could enhance overall treatment efficacy. Indeed, some studies have shown that the combination of NLRP3 inhibitors with checkpoint inhibitors could improve therapeutic efficacy by modifying the immune surveillance in the tumor microenvironment and tumor cells. Recent studies have also identified novel biomarkers for various cancers, and understanding how NLRP3 inhibition alters these biomarker responses could help develop better therapeutic approaches. Further, the use of next-generation sequencing and omics could help to identify better drugs to inhibit NLRP3 inflammasome, and could also identify possible prognosis markers for cancer.

In addition, the current understanding of how the existing NLRP3 inhibitors exert any side-effects is important. This will help to develop specific inhibitors with lower side-effects when treating cancer patients. Generally, the inflammasome-mediated immune and inflammatory responses contribute to immune surveillance, pathogen recognition, and cell death; inhibiting it might imbalance the immune responses and increase the infection risk or other adverse effects. Therefore, additional safety-profile studies will be needed to ensure better cancer therapy with the NLRP3 inhibitors to control any unintended side-effects. Developing such new inhibitors with greater specificity and fewer off-target effects could improve their therapeutic potential. Although the potential use of NLRP3 inflammasome inhibitors in cancer therapy has been developing for the last decade, the therapeutic approach should be directed to the inflammasome’s specific role in different cancer types. This individualized treatment of each cancer type should be adjusted to the specific role of inflammasome in cancer progression or inhibition. Thus, by understanding how the NLRP3 inflammasome either promotes or inhibits cancer growth in various tissues, we can develop more precise and effective treatments, using NLRP3 inhibitors alone or in combination with other immunotherapy agents.

Thus, recent studies suggest that inhibition of NLRP3 inflammasome-mediated immune and inflammatory responses provides innovative strategies to control cancer growth and metastasis. Since NLRP3 might be involved in the tumor progression as well as pyroptosis of cancer cells, ongoing research should further explore the effects of NLRP3 inhibition on specific cancer types and stages, to understand how these inhibitors alter carcinogenic processes at different levels. Additionally, pre-clinical studies using advanced technologies could help better prognosis and treatment strategies with NLRP3 inhibitors alone or in combination with other therapeutic approaches, especially immunotherapies.

## Figures and Tables

**Figure 1 ijms-25-13689-f001:**
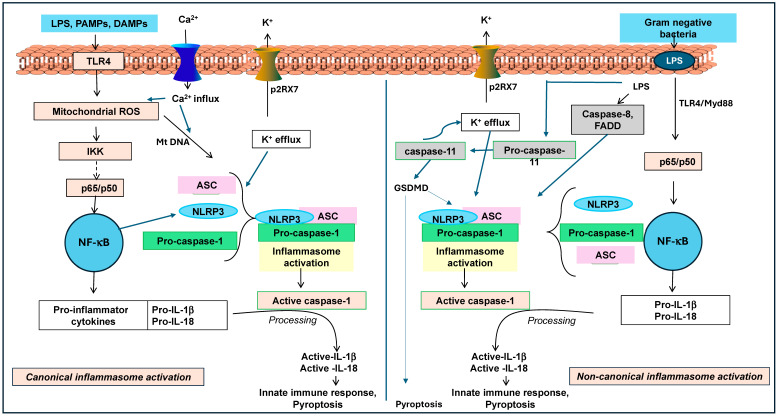
Canonical and non-canonical activation of NLRP3 inflammasome. The canonical pathway involves the activation of NLRP3 inflammasomes through signals such as mitochondrial ROS, calcium influx, and potassium efflux, leading to NF-κB activation and the production of pro-inflammatory cytokines (IL-1β and IL-18). This pathway ultimately activates caspase-1, resulting in cytokine release and pyroptosis. The non-canonical pathway involving LPS from Gram-negative bacteria triggers caspase-11, which indirectly activates NLRP3, leading to similar inflammasome responses, cytokine production, and pyroptosis. Both pathways generally play critical roles in innate immunity and inflammation.

**Figure 2 ijms-25-13689-f002:**
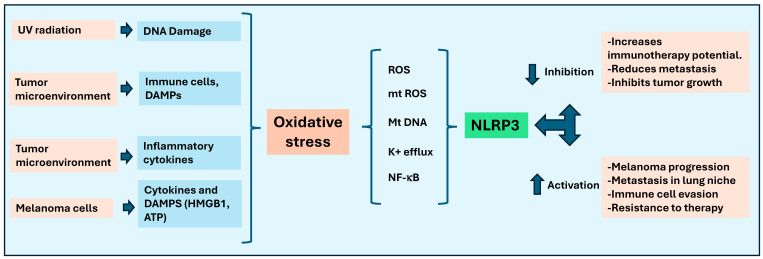
Significance of NLRP3 inflammasome in the melanoma progression. Various factors such as UV radiation, tumor microenvironment, and melanoma cells contribute to oxidative stress and cause DNA damage, immune cell activation, and cytokine release. Oxidative stress, in turn, triggers reactive oxygen species (ROS), mitochondrial DNA damage, potassium efflux, and NF-κB activation, which influence the activation of the NLRP3 inflammasome. NLRP3 activation promotes melanoma progression, metastasis, immune evasion, and therapy resistance. In contrast, inhibition of NLRP3 could enhance immunotherapy, inhibit tumor growth, and reduce metastasis.

**Figure 3 ijms-25-13689-f003:**
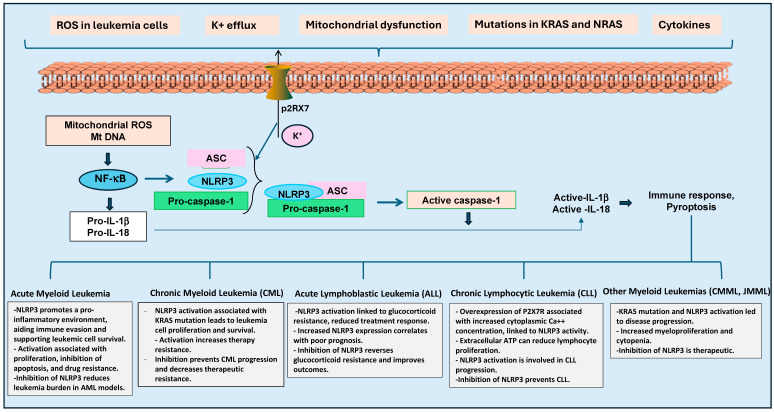
Significance of NLRP3 inflammasome in leukemias. Oxidative stress and mitochondrial dysfunction in leukemia cells could activate NLRP3 inflammasomes through mitochondrial ROS, potassium efflux, and NF-κB signaling pathways. NLRP3 activation leads to the generation of active IL-1β and IL-18 cytokines, and could cause pyroptosis. Further, NLRP3 activation plays various roles in different leukemias. For example, in Acute Myeloid Leukemia (AML), NLRP3 promotes immune evasion and survival, while inhibition reduces the disease burden. In Chronic Myeloid Leukemia (CML), NLRP3 is linked to KRAS mutations and therapy resistance. In Acute Lymphoblastic Leukemia (ALL), NLRP3 activation is correlated with glucocorticoid resistance, and in Chronic Lymphocytic Leukemia (CLL), P2X7R overexpression leads to increased NLRP3.

**Figure 4 ijms-25-13689-f004:**
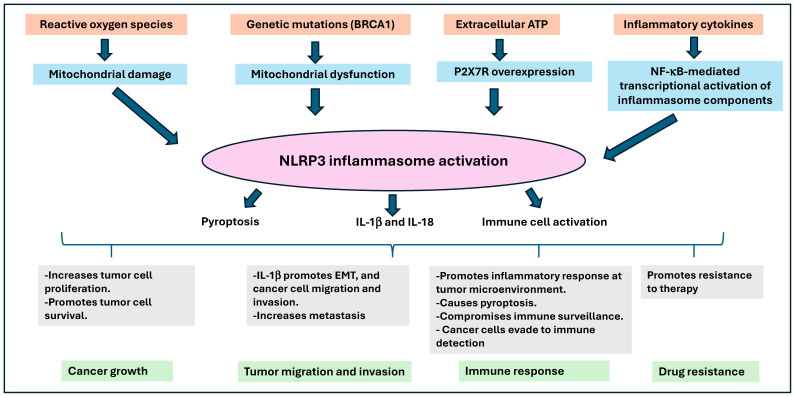
Role of NLRP3 inflammasome in breast cancer growth and spread. Several factors such as reactive oxygen species (ROS)-induced mitochondrial damage, BRCA1-associated genetic mutations causing mitochondrial dysfunction, extracellular ATP leading to P2X7R overexpression, and inflammatory cytokines that activate NF-κB-mediated inflammasome components could lead to activation of NLRP3 inflammasome. NLRP3-mediated release of IL-1β and IL-18 promotes cancer cell proliferation, survival, migration, immune evasion, and resistance to therapy. Further, the outcomes also include increased tumor growth, metastasis, compromised immune surveillance, and drug resistance.

**Figure 5 ijms-25-13689-f005:**
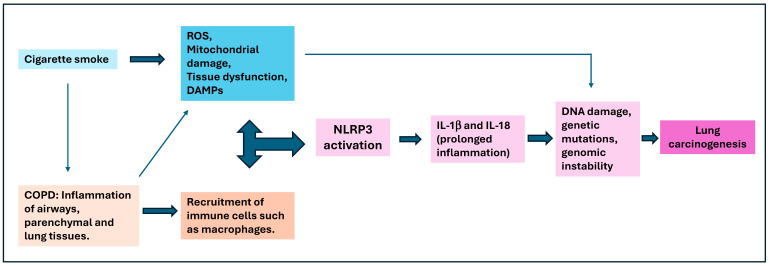
Significance of NLRP3 inflammasome activation in cigarette smoke and COPD-induced lung cancer development. Cigarette smoke leads to reactive oxygen species (ROS), mitochondrial damage, and tissue dysfunction, activating the NLRP3 inflammasome. The release of pro-inflammatory cytokines IL-1β and IL-18 contributes to prolonged inflammation. COPD-induced inflammation and immune cell recruitment, such as macrophages, further amplify this process. The persistent inflammation and oxidative stress promote DNA damage, genetic mutations, and genomic instability, ultimately leading to lung carcinogenesis.

**Figure 6 ijms-25-13689-f006:**
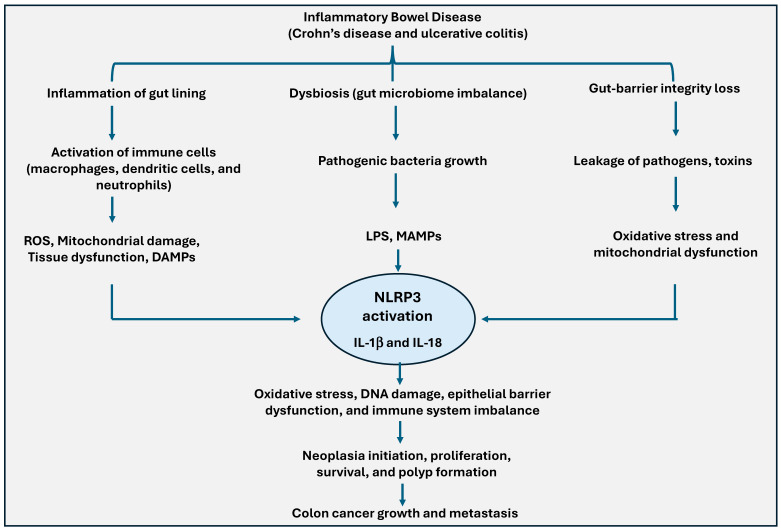
Role of NLRP3 inflammasome activation in promoting colon cancer development. During inflammatory bowel disease (IBD), such as Crohn’s disease and ulcerative colitis, the inflammation of the gut lining activates immune cells, causing oxidative stress, mitochondrial damage, and the release of DAMPs. Gut microbiome imbalance (dysbiosis) leads to pathogenic bacterial growth and loss of gut-barrier integrity, allowing pathogen and toxin leakage, further driving oxidative stress. These pathways cause NLRP3 activation and trigger IL-1β and IL-18 release. Increased inflammasome response results in immune system imbalance, epithelial-barrier dysfunction, and neoplasia initiation, ultimately contributing to colon cancer growth and metastasis.

**Table 1 ijms-25-13689-t001:** Significance of NLRP3 inflammasome in gastric and pancreatic cancers.

Cancer Type	Role of NLRP3	Citations
Gastric cancer	*H. pylori* infection activates NLRP3 and releases IL-1β and IL-18, leading to chronic inflammation.	[156,157]
	High NLRP3 expression observed in gastric adenocarcinoma serves as a potential prognostic biomarker.	[148,158,159]
	*H. pylori*-generated ROS enhances NLRP3 expression and alters macrophage polarization.	[160]
	Inhibition of NLRP3 reduces CagA-mediated gastric cancer cell migration.	[161]
	IL-1β regulation by NLRP3 linked to gastric cancer development.	[162,163]
	Mitochondrial oxidative damage from *H. pylori* influences NLRP3 activity.	[165,166,167]
	Paclitaxel prevents NLRP3-mediated migration and invasion.	[170]
	Long non-coding RNA ADAMTS9-AS2 inhibits gastric cancer through the miR-223-3p/NLRP3 axis.	[171]
	Gut microbiota disruptions by *H. pylori* enhance NLRP3-mediated inflammatory response.	[172,173]
Pancreatic Cancer	NLRP3 pathway activation induces pancreatitis and chronic inflammation.	[174,175,176,177]
	IL-1β promotes pancreatic injury and metastasis.	[178,179,180,181]
	Genetic polymorphisms in NLRP3 are linked to increased pancreatic cancer risk.	[182]
	NLRP3 promotes platelet activation and aggregation, which contributes to tumor growth.	[183]
	Downregulation of NLRP3 inhibits cancer cell proliferation, suggesting a tumor-suppressive role.	[184]
	NLRP3 is linked to islet damage and inflammation.	[185,186,187,188]

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
