# Peer review of "Therapeutic Significance of NLRP3 Inflammasome in Cancer: Friend or Foe?"

_ijms, 2024, doi:10.3390/ijms252413689_

Round 1
Reviewer 1 Report
Comments and Suggestions for Authors
Dear Authors
I would suggest to incorporate in the Introduction a few lines about the relationship between cancer and inflammation.
References
Dvorak HF. Tumors: wounds that do not heal. Similarities between tumor stroma generation and wound healing. N Engl J Med. 1986 Dec 25;315(26):1650-9. doi: 10.1056/NEJM198612253152606. PMID: 3537791
and
Dvorak, H. F. (2019, September). Tumors: wounds that do not heal—a historical perspective with a focus on the fundamental roles of increased vascular permeability and clotting. In Seminars in thrombosis and hemostasis (Vol. 45, No. 06, pp. 576-592). Thieme Medical Publishers
This would give a short historical background to the paper.
Inflammasome requires a wider explanation for the non-expert reader.
Under the heading of melanoma, please include the type of UV radiation that represents a risk for melanoma. There are three different types of UV according to wave length.
In the risk factors for melanoma please include UV beds.
There are many abbreviations without giving the full name of the acronym the first time they are mentioned. For example NLPR3, MAPK, NF-kB, IL-1, PAMPs, DAMPs, IFN, MCP-1, MIF-1, EMT, UV, HMGB1, AHR, CASP1, CDDO-2P, P2X7R, iFGFR1, MMP3, SESN2 , COPD, LPS, AOM/DSS, CagA, NALP3, and many others.
Please give some details about zerumbone. For example you can add that Zerumbone is an orally active natural cyclic sesquiterpene and can be isolated from Zingiber zerumbet. Zerumbone has anti-proliferative, anti-inflammation, anti-cancer, anti-bacterial and anti-mutagenic activity.
Reference: Sulaiman M R, et al. Anti-inflammatory effect of zerumbone on acute and chronic inflammation models in mice [J]. Fitoterapia, 2010, 81(7): 855-858.
In the section of breast cancer, inflammatory breast cancer should be included and discussed.
Line 334 IL-1b
Lynch syndrome should be called hereditary non-polyposis colon cancer, which is the updated name of the syndrome.
Line 426 Gastric cancer, also called stomach cancer, Please remove the also called stomach cancer.
The Table should show the references.
Author Response
I would suggest to incorporate in the Introduction a few lines about the relationship between cancer and inflammation.
References
Dvorak HF. Tumors: wounds that do not heal. Similarities between tumor stroma generation and wound healing. N Engl J Med. 1986 Dec 25;315(26):1650-9. doi: 10.1056/NEJM198612253152606. PMID: 3537791 and Dvorak, H. F. (2019, September). Tumors: wounds that do not heal—a historical perspective with a focus on the fundamental roles of increased vascular permeability and clotting. In Seminars in thrombosis and hemostasis (Vol. 45, No. 06, pp. 576-592). Thieme Medical Publishers
This would give a short historical background to the paper.
As suggested, we have expanded the introduction showing the relationship between cancer and inflammation. However, the suggested references are not directly related to inflammation and cancer. We are sorry for not including these references in the revised manuscript.
Inflammasome requires a wider explanation for the non-expert reader.
As suggested, we have provided additional information.
Under the heading of melanoma, please include the type of UV radiation that represents a risk for melanoma. There are three different types of UV according to wave length.
As suggested, we have included this information.
In the risk factors for melanoma please include UV beds.
As suggested, we have included this information.
There are many abbreviations without giving the full name of the acronym the first time they are mentioned. For example NLPR3, MAPK, NF-kB, IL-1, PAMPs, DAMPs, IFN, MCP-1, MIF-1, EMT, UV, HMGB1, AHR, CASP1, CDDO-2P, P2X7R, iFGFR1, MMP3, SESN2 , COPD, LPS, AOM/DSS, CagA, NALP3, and many others.
As suggested, we have carefully defined all the abbreviation at the first use.
Please give some details about zerumbone. For example you can add that Zerumbone is an orally active natural cyclic sesquiterpene and can be isolated from Zingiber zerumbet. Zerumbone has anti-proliferative, anti-inflammation, anti-cancer, anti-bacterial and anti-mutagenic activity.
As suggested, we have included this information.
Reference: Sulaiman M R, et al. Anti-inflammatory effect of zerumbone on acute and chronic inflammation models in mice [J]. Fitoterapia, 2010, 81(7): 855-858.
In the section of breast cancer, inflammatory breast cancer should be included and discussed.
As suggested, we have included this information.
Line 334 IL-1b, Sorry for the mistake, we replaced underscore with hyphen.
Lynch syndrome should be called hereditary non-polyposis colon cancer, which is the updated name of the syndrome.
As suggested, we have indicated this information.
Line 426 Gastric cancer, also called stomach cancer, Please remove the also called stomach cancer.
As suggested, we have deleted stomach cancer ward.
The Table should show the references.
As suggested, citations were included in the table.

Reviewer 2 Report
Comments and Suggestions for Authors
In this review article, the authors discussed the role of the NLRP3 inflammasome in
pathophysiology of various types of cancer. Elucidation of this role may be helpful in the development of new anti-cancer therapeutic strategies. NRLP3 belongs to the pattern recognition receptor protein family (PRR) and is important in the innate cellular immune response. Depending on the type of cancer, it may play a protective or harmful role.
The role of NRLP3 in such cancers as HNCs, gynecological cancers, prostate cancer, colon cancer, gastric and pancreatic cancer, breast cancer, lung cancer and leukemia has been described in great detail.
The discussion of the issue is based on a critical review of the available scientific literature. The authors used 253 references. The authors rightly postulate the need for further research on NLRP3 inhibitors and their use in cancer therapy, paying attention to the side effect.
I consider this review to be a very valuable item in the scientific literature.
Author Response
In this review article, the authors discussed the role of the NLRP3 inflammasome in pathophysiology of various types of cancer. Elucidation of this role may be helpful in the development of new anti-cancer therapeutic strategies. NRLP3 belongs to the pattern recognition receptor protein family (PRR) and is important in the innate cellular immune response. Depending on the type of cancer, it may play a protective or harmful role.
The role of NRLP3 in such cancers as HNCs, gynecological cancers, prostate cancer, colon cancer, gastric and pancreatic cancer, breast cancer, lung cancer and leukemia has been described in great detail.
The discussion of the issue is based on a critical review of the available scientific literature. The authors used 253 references. The authors rightly postulate the need for further research on NLRP3 inhibitors and their use in cancer therapy, paying attention to the side effect.
I consider this review to be a very valuable item in the scientific literature.
We thank the reviewer for considering our article valuable.
Reviewer 3 Report
Comments and Suggestions for Authors
The manuscript is very interesting. The authors review the activity of inflammasomes in different types of cancer and their role in cancer progression showing that it differed in different cancer types and their binary function in promoting tumor growth and metastasis. It shows also that immunotherapy in each cancer type can have different effects. The authors focus on the prospect use of inhibitors of NLRP3 in inflammasome in cancer treatment, and that the treatment of each cancer type should be adjusted to the specific role of inflammasome in cancer progression or inhibition.
Author Response
The manuscript is very interesting. The authors review the activity of inflammasomes in different types of cancer and their role in cancer progression showing that it differed in different cancer types and their binary function in promoting tumor growth and metastasis. It shows also that immunotherapy in each cancer type can have different effects.
The authors focus on the prospect use of inhibitors of NLRP3 in inflammasome in cancer treatment, and that the treatment of each cancer type should be adjusted to the specific role of inflammasome in cancer progression or inhibition.
As suggested, we have included this information Prospectives Section.